# A randomized and open-label phase II trial reports the efficacy of neoadjuvant lobaplatin in breast cancer

Xiujuan Wu[1], Peng Tang[1], Shifei Li[1], Shushu Wang[1], Yueyang Liang[1], Ling Zhong[1], Lin Ren[1], Ting Zhang[1] & Yi Zhang[1]

Currently, one sixth of triple-negative breast cancer (TNBC) patients who receive docetaxel (T) and epirubicin (E) as neoadjuvant chemotherapy achieve a pathologic complete response (pCR). This study evaluates the impact of adding lobaplatin (L) to the TE regimen. Here, we show data from 125 patients (63 TE and 62 TEL patients). Four patients did not complete all the cycles. Two-sided $P$ values show that the addition of L (38.7% vs. 12.7%, $P = 0.001$) significantly increases the rate of pCR in the breast and the axilla (TpCR) and the overall response rate (ORR; 93.5% vs. 73.0%, $P = 0.003$). The occurrence of grade 3–4 anemia and thrombocytopenia is higher in the TEL group (52.5% vs. 10.0% and 34.4% vs. 1.7% respectively). These results demonstrate that the addition of L to the TE regimen as neoadjuvant chemotherapy improves the TpCR and the ORR rates of TNBC but with increased side effects.

[1] Breast Disease Center, Southwest Hospital, Third Military Medical University, Chongqing, 400038, China. Correspondence and requests for materials should be addressed to Y.Z. (email: zhangyi1489@sina.com)

Triple-negative breast cancers (TNBC) are invasive breast cancers with absent or minimal expression of the estrogen receptor (ER), progesterone receptor (PR), and human epidermal growth factor receptor 2 (HER2) and account for ~10–15% of all breast cancers[1]. A worse prognosis has been reported in TNBC patients compared with hormone receptor or HER2-positive breast cancer patients[2]. Endocrine therapy and HER2 molecular-targeted therapy do not appear to have benefits in TNBC patients, and adjuvant chemotherapy is recommended as a standard treatment for TNBC patients. A previous study reported that patients with TNBC have increased pCR rates compared with non-TNBC patients when given neoadjuvant chemotherapy[3]. Some meta-analyses have evaluated the impact of supplementing local therapy with chemotherapy for TNBC patients, and the results have shown that polychemotherapy leads to significant reductions in breast cancer recurrence and mortality[4]. The improvements in adjuvant chemotherapy have a greater impact on the prognosis of TNBC than on hormone receptor-positive breast cancer.

The combination of docetaxel (T) and epirubicin (E) is a widely used treatment regimen. Through a phase III trial (ABCSG-24), Steger et al. confirmed the efficacy of neoadjuvant chemotherapy with TE for early breast cancer[5,6]. TNBC patients exhibit a higher risk of recurrence, equal to 10–15% per year during the first several years after initial surgery, than hormone receptor-positive breast cancer patients, who show recurrence rates of 3–5%[7]. A new chemotherapeutic option, platinum-based neoadjuvant chemotherapy combined with TE, is currently being evaluated for TNBC.

Platinum agents induce DNA-crosslinking events and subsequent apoptosis, leading to cell death[8]. Some studies of platinum-based neoadjuvant polychemotherapy have reported rates of pCR in the breast and axilla (TpCR) as high as 31%[9], whereas other studies have shown that single-agent platinum therapy yields few pCRs (22%) in TNBC patients[10]. Platinum-based compounds (cisplatin or carboplatin) are among the most widely used and effective agents. However, the administration of cisplatin is frequently limited by several severe toxicities, including gastrointestinal toxicity, nephrotoxicity and neurotoxicity. Although two randomized phase II neoadjuvant trials have confirmed that platinum-based (carboplatin) neoadjuvant chemotherapy significantly increases the pCR rates (53.2% vs. 36.9% and 60% vs. 44%, respectively) in TNBC[11,12], neither an increase in the rate of breast conservation nor long-term overall survival was reported, and several severe toxicities were detected. The identification of another platinum-based chemotherapy drug with reduced toxicity and a better therapeutic index is essential. Lobaplatin [1,2-diamminomethylcyclobutane-platinum (II) lactate] represents the third-generation platinum anticancer drugs, which show strong anticancer activity, low toxicity, and higher solubility and stability in water[13]. The mechanism of action of lobaplatin is similar to that of other platinum-based compounds. In vitro experimental studies and a variety of preclinical tumor models have demonstrated that lobaplatin inhibits cell proliferation and shows promising anti-neoplastic effects[14,15] due to the formation of DNA-drug adducts and influences on c-myc gene expression, which result in cell cycle arrest and changes in the expression of numerous proteins. In fact, lobaplatin has been shown to exert obvious effects in the treatment of various tumors in China. Deng et al.[16] indicated that the chemotherapy regimen of lobaplatin and pemetrexed is an effective treatment for metastatic breast cancer patients, but the doses of chemotherapy should be further modified. Another study of human non-small-cell lung cancer reported that the combination of lobaplatin with paclitaxel shows enhanced activity compared with cisplatin combined with paclitaxel[17]. Lobaplatin has also been shown to

exhibit therapeutic efficacy in the treatment of hepatic cancer[18,19], with a total effective rate of 67.6%.

It is currently unclear whether a regimen of neoadjuvant polychemotherapy with TE would benefit from the addition of lobaplatin to improve the pCR rate in TNBC patients. Here, we report a randomized controlled trial (RCT) designed to examine the impact of a TE+lobaplatin (TEL) regimen on the clinical treatment of TNBC in terms of TpCR, overall response rate (ORR) and toxicity. We show that the addition of lobaplatin significantly increases TpCR and ORR and that improvement in toxicities are observed after symptomatic treatment. These results demonstrate that the TEL neoadjuvant chemotherapy regimen exhibits short-term efficacy with reversible side-effects.

## Results

**Patient characteristics**. As Fig. 1 shows, 128 patients were enrolled in this study between January 2014 and February 2017. Three of these patients had not begun a treatment protocol. The other 125 patients were randomly allocated to groups 1 or 2 (63 in the patients the TE group (median tumor size 2.8 cm (range 0.7–6.00 cm)) and 62 patients in the TEL group (median tumor size 2.97 cm (range 0.94–7.00 cm)). The characteristics of the 125 treated patients are listed in Table 1. Their ages ranged from 33 to 70 years (with a median age of 47 years), and the age, clinical stage, histological type, maximum tumor diameter, and lymphatic metastasis of the patients were evenly distributed between the two groups.

Of the patients who started the treatment, one patient stopped due to drug toxicity, two patients stopped due progressive disease (PD), and one patient refused treatment. A total of 121 patients completed their respective treatment cycles (Fig. 1).

**Clinical efficacy**. After neoadjuvant chemotherapy, the patients treated with the TEL chemotherapy regimen showed a TpCR rate of 38.7% (24/62), whereas the patients who were treated with the TE regimen showed a TpCR rate of 12.7% (8/63) (odds ratio (OR), 4.342, 95% CI 1.764–10.687; $P = 0.001$; Fig. 2). The results of the TE regimen with added lobaplatin are summarized in Table 2. The TEL group included 29 patients with CR, 29 with PR, three with SD, and none with PD; their ORR, which included complete response (CR) and partial response (PR), was 93.5% (58/62). The TE group consisted of 20 patients with CR, 26 patients with PR, 14 patients with SD, and 2 patients with PD and presented an ORR of 73.0% (46/63). The patients who underwent the TEL regimen had the highest ORR, as shown in Fig. 2 (OR, 5.359, 95% CI 1.687–17.024, $P = 0.004$).

**Adverse reactions**. All the patients included in the clinical study underwent toxicity assessments. Blood tests were performed before and after every chemotherapy cycle and as necessary. The most common adverse events are shown in Tables 3 and 4. Serious adverse events were defined as any unexpected grade 3 +toxicity or toxicity requiring treatment at the hospital. Compared with the patients assigned to the non-lobaplatin-containing regimens, those assigned to the TEL regimen were more likely to develop grade 3–4 thrombocytopenia (34.4% vs. 1.7%, $P < 0.001$) and anemia (52.5% vs. 10%, $P < 0.001$). The incidences of leukopenia and neutropenia were not significantly different between the TEL and TE groups. Digestive reactions, including vomiting (3/61 vs. 2/60 for the TEL and TE groups, respectively) and diarrhea (2/61 vs. 2/60, respectively), occurred at similar rates. There was one case of subcutaneous hemorrhage and one case of phlebitis in the TEL group, but neither of these toxicities were observed in the TE group. No obvious liver or kidney toxicity and no neurotoxicity were reported, and no patients died due to

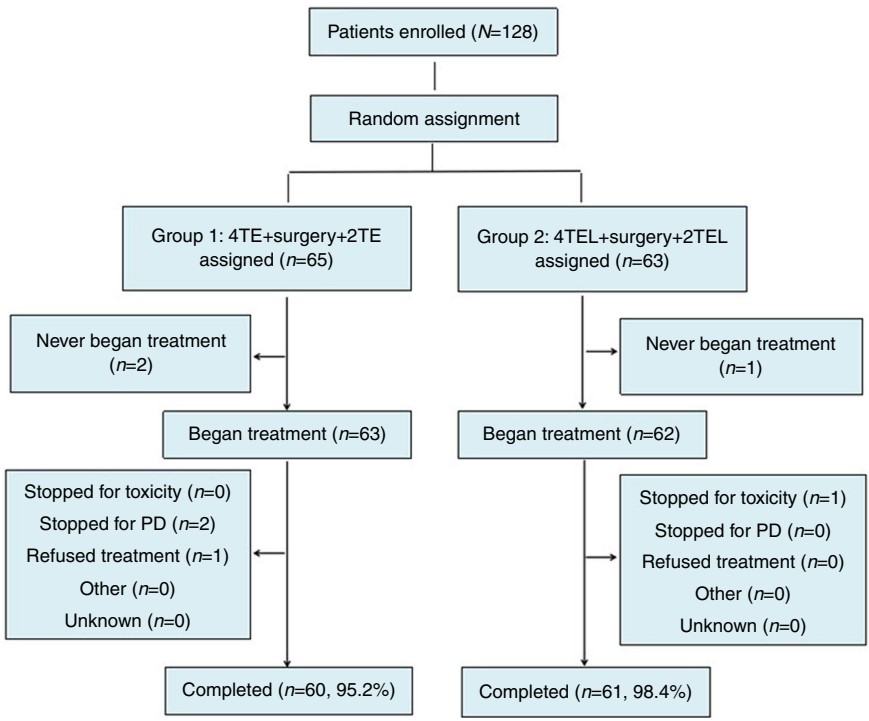

**Fig. 1** CONSORT diagram. TE docetaxel and epirubicin, TEL docetaxel, epirubicin, and lobaplatin

| Table 1 Patient demographic and clinical characteristics | | | | |
|---|---|---|---|---|
| **Characteristics** | **Total patients (began treatment) (N = 125) No. (%)** | **Group 1 TE (N = 63) %** | **Group 2 TEL (N = 62) %** | **Rank-sum tests (TE/TEL) P** |
| *Age (years)* | | | | P = 0.100 |
| <40 | 17 (14) | 13 | 15 | |
| 40–59 | 96 (77) | 71 | 82 | |
| ≥60 | 12 (9) | 16 | 3 | |
| *Clinical stage (n)* | | | | P = 0.091 |
| I | 11 (9) | 10 | 8 | |
| II | 83 (67) | 73 | 60 | |
| III | 31 (24) | 17 | 32 | |
| IV | 0 (0) | 0 | 0 | |
| *Histological type (n)* | | | | P = 0.762 |
| Invasive ductal cancer | 116 (93) | 92 | 94 | |
| Invasive lobular cancer | 3 (2) | 3 | 1 | |
| Invasive special cancer | 6 (5) | 5 | 5 | |
| *N stage (n)* | | | | P = 0.989 |
| 0 | 50 (40) | 36 | 43 | |
| 1 | 48 (38) | 46 | 31 | |
| 2 | 10 (8) | 8 | 8 | |
| 3 | 17 (14) | 10 | 18 | |
| *Tumor size (cm)* | | | | P = 0.643 |
| 0.5–2 | 38 (30) | 33 | 27 | |
| 2.1–3.5 | 51 (41) | 38 | 44 | |
| 3.6–5 | 32 (26) | 25 | 26 | |
| 5.1–7 | 4 (3) | 4 | 3 | |
| *TE docetaxel and epirubicin, TEL docetaxel, epirubicin, and lobaplatin* | | | | |

adverse reactions. All the patients showed improvement after receiving symptomatic treatment.

**Survival analysis**. Because the early trial was completed recently, the follow-up times were not extensive at the time of this writing. Thus, in the current study, we report the recurrence and metastasis rates as preliminary survival information. The occurrence of recurrence and metastasis was significantly greater in the TE group (nine cases) than in the TEL group (two cases; log-rank $P = 0.028$). The HR for the development of recurrence and metastasis was 4.755 (95% CI 1.026–22.035) for the in the TE group compared with those in the TEL group (Fig. 3).

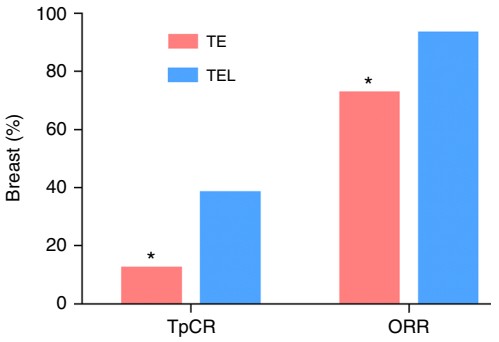

**Fig. 2** Pathologic complete response in the breast and the axilla (TpCR) and overall response rate (ORR) of TNBC patients after 4 cycles of neoadjuvant chemotherapy. TE docetaxel and epirubicin, TEL docetaxel, epirubicin, and lobaplatin. *indicates statistically significant differences between the two groups ($P < 0.05$)

### Table 2 Clinical response to chemotherapy

| Response | Group 1 TE N (%) | Group 2 TEL N (%) |
|---|---|---|
| CR | 20 (31.7) | 29 (46.8) |
| PR | 26 (41.3) | 29 (46.8) |
| SD | 14 (22.2) | 3 (4.8) |
| PD | 2 (3.2) | 0 (0) |
| Total | 63 (100) | 62 (100) |

*TE* docetaxel and epirubicin, *TEL* docetaxel, epirubicin, and lobaplatin, *CR* complete response, *PR* partial response, *SD* stable disease, *PD* progression of disease

### Table 3 Hematological toxicities

| | Group 1 TE (60) N (%) | Group 2 TEL (61) N (%) |
|---|---|---|
| *Leukopenia* | | |
| I–II | 26 (43.3) | 29 (47.5) |
| III–IV | 14 (23.3) | 20 (32.8) |
| *Neutropenia* | | |
| I–II | 19 (31.7) | 18 (29.5) |
| III–IV | 14 (23.3) | 24 (39.3) |
| *Anemia* | | |
| I–II | 41 (68.3) | 26 (42.6) |
| III–IV | 6 (10.0) | 32 (52.5)* |
| *Thrombocytopenia* | | |
| I–II | 7 (11.7) | 11 (18.0) |
| III–IV | 1 (1.7) | 21 (34.4)* |

*TE* docetaxel and epirubicin, *TEL* docetaxel, epirubicin, and lobaplatin
* Indicates significant difference in incidence compared with the other treatment arm ($P < 0.05$)

### Table 4 Grade III–IV treatment-related non-hematologic toxicities

| | Group 1 TE (60) N | Group 2 TEL (61) N |
|---|---|---|
| Vomiting | 2 | 3 |
| Diarrhea | 2 | 2 |
| Subcutaneous hemorrhage | 0 | 1 |
| Phlebitis | 0 | 1 |
| Liver and kidney toxicity | 0 | 0 |
| Neurotoxicity | 0 | 0 |

*TE* docetaxel and epirubicin, *TEL* docetaxel, epirubicin, and lobaplatin

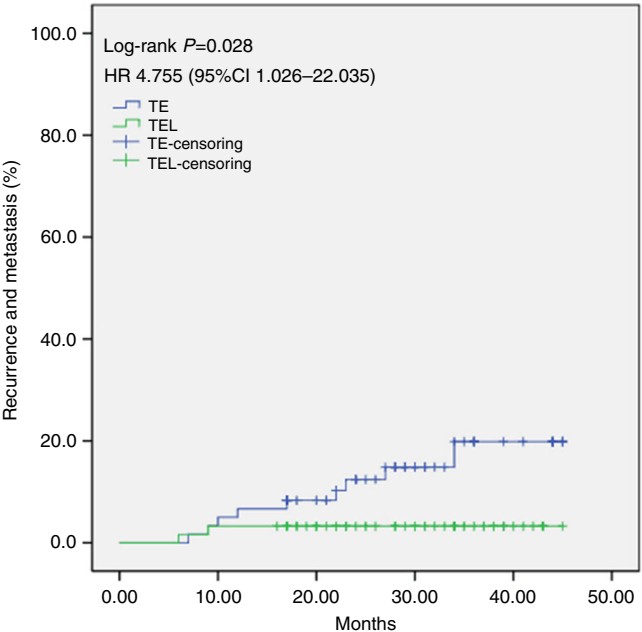

**Fig. 3** Cumulative incidence of breast tumor recurrence and metastasis. HR hazard ratio

## Discussion

The current phase II clinical trial is the first RCT to investigate the efficacy and toxicity of lobaplatin combined with docetaxel and epirubicin for TNBC. Our results demonstrate that the combination regimen exhibited short-term efficacy with mild and tolerable toxicities. The randomized trial indicates that the addition of lobaplatin to the docetaxel and epirubicin regimen resulted in a significant increase in TpCR. As observed in previous studies, myelosuppression is the major dose-limiting toxicity of lobaplatin[18,20], and this effect typically manifests as thrombocytopenia, anemia, and leukopenia ~2 weeks after drug administration.

The maximum tolerated dose of lobaplatin when used in combination therapies is not clear in Asian populations. In Germany, phase I/II clinical trials have established a recommended dose of 50 mg m$^{-2}$ for single-agent lobaplatin therapy in Western subjects. However, dose recommendations based on studies of Western populations are not always applicable to Eastern patients due to potentially different tolerances of the same doses of chemotherapy. Additionally, the dose of lobaplatin in combination therapies has not been previously studied. In China, recommendations of a dose of 50 mg m$^{-2}$ for single-agent lobaplatin chemotherapy and a dose of 30 mg m$^{-2}$ for combination therapies have gained consensus. A phase I clinical trial of increasing doses of lobaplatin showed dose-limiting toxicity in 10 patients with a dose of 35 mg m$^{-2}$, but no adverse events were observed with 30 mg m$^{-2}$. In addition, the therapeutic effect of combined agents is superior to that of single-agent chemotherapy[21]. Docetaxel and epirubicin are broad-spectrum antitumor drugs that can effectively treat a variety of malignant solid tumors. The doses selected to be administered were close to the median doses reported in a study that evaluated docetaxel dose in breast cancer[22] and in a review of epirubicin[23]. In one phase III trial, the time to disease progression or overall survival with docetaxel doses ranging from 60 to 100 mg m$^{-2}$ was similar for the adjuvant treatment of intention-to-treat patients with advanced breast cancer, while significant aggravation of hematologic toxicity occurred in the highest docetaxel dose group, with a 93.4% incidence of grade 3 to 4 neutropenia[22]. In addition, in an

analysis of the FinHer trial data, there were no significant differences in the baseline characteristics of the patients or the tumors, the numbers of distant recurrences or deaths between the groups receiving 80 mg m$^{-2}$ and 100 mg m$^{-2}$ of docetaxel[24]. When the drug dosage increased, there was more myelotoxicity without necessarily a higher response rate. Hence, our clinical trial used a lobaplatin dose of 30 mg m$^{-2}$ combined with docetaxel 75 mg m$^{-2}$ and epirubicin 80 mg m$^{-2}$ as a neoadjuvant chemotherapy regimen for TNBC patients.

Increasing evidence has revealed that pCR is the most powerful predictive factor for the effectiveness of neoadjuvant chemotherapy and prognosis in TNBC patients. TNBC patients with known pCR have a better prognosis than those with residual tumor[25]. Liedtke et al. noted that TNBC patients with pCR have excellent survival[3]. A meta-analysis[26] published in Lancet in 2014 confirmed that patients with TNBC who achieved pCR demonstrated superior event-free survival (HR, 0.24; 95% CI, 0.18–0.33) and overall survival (HR, 0.16; 95% CI, 0.11–0.25) than those who did not achieve a pCR. Keam et al.[27] reported that the pCR rates in both the breast and the axillary lymph nodes for 47 TNBC patients who received docetaxel and doxorubicin combination therapy was 17%, and another study discerned that patients with locally advanced TNBC showed an unfavorable pCR rate in response to the combination of epirubicin and taxanes[28]. Platinum-based chemotherapy has been proposed as another regimen for TNBC. Several studies have reported that the pCR rates of TNBC patients increased to 22% with the use of cisplatin alone as an adjuvant chemotherapy[10], and this rate could reach 40–67% through the combination of cisplatin or carboplatin with anthracycline or taxanes[29,30]. Our study is the first RCT showing that lobaplatin combined with docetaxel and epirubicin significantly improves the TpCR rate to 38.7% compared with non-lobaplatin-based chemotherapy. However, the pCR rate achieved in our study is lower than that reported previously[30]. One possible reason for this discrepancy is the different cycles of neoadjuvant chemotherapy: we used a total of four cycles of chemotherapy every 3 weeks, whereas another study used eight preoperative weekly cycles.

The ORR, including CR and PR, is another metric used to evaluate the effect of neoadjuvant chemotherapy. The ORR can effectively reflect the sensitivity of chemotherapy agents and predict the patient's prognosis[31,32]. Deng et al.[16] reported that the ORR rate of patients with metastatic breast cancer who received lobaplatin was as low as 15.8%. In our study, the ORRs obtained with the TEL and TE regimens reached 93.5% (29CR + 29PR) and 73.0% (20CR + 26PR, $P < 0.05$), respectively; these results are superior to those reported in the above-mentioned studies, possibly because TNBC is more sensitive to chemotherapy than other subtypes of breast cancer. Moreover, lobaplatin can significantly increase the ORR rate of TNBC.

The evaluation and gradation of anticancer toxic drug reactions were based on the Common Terminology Criteria for Adverse Events, v3.0[33]. Myelosuppression is the main toxicity of lobaplatin observed in previous trials[34–36], and thrombocytopenia is the most common dose-limiting toxicity. In multiple clinical trials, the incidence of grade 3–4 thrombocytopenia ranged from 26.0–72.7% with a dose of 50 mg m$^{-2}$ [37,38], whereas an incidence rate of 21.3–31.8% was observed with a dose of 35 mg m$^{-2}$ [16,35]. Therefore, we considered the incidence of thrombocytopenia to be primarily associated with the lobaplatin dose, and the lobaplatin dose was reduced based on observations of thrombocytopenia. In the current study, the incidence of grade 3–4 thrombocytopenia in the lobaplatin-based chemotherapy group was significantly higher than that observed in the non-lobaplatin-based chemotherapy group (34.4% vs. 1.7%, $P < 0.05$), and interleukin-11 treatment enabled a return to normal after

thrombocytopenia in all but one patient, who stopped the treatment due to the toxicity. Moreover, leukopenia, neutropenia and anemia have also been reported in several clinical trials. Our study observed a higher incidence of grade 3–4 anemia in the TEL group (52.5%) compared with the TE group (10.0%), consistent with the early results of other studies (3–12%)[16,36], which suggests that severe myelosuppression is an effect of the TEL chemotherapy regimen. Myelosuppression might be a result of the cytotoxicity of lobaplatin or the interaction between lobaplatin and docetaxel. In addition, anemia was improved by treatment with erythrocyte growth factor, and no patients required red blood cell transfusions. Curiously, the incidence of leukopenia and neutropenia did not differ significantly between the two groups, which might be attributed to the prophylactic use of granulocyte colony-stimulating factor. Phlebitis, subcutaneous hemorrhage, digestive tract reaction, and hepatic and renal toxicity should be monitored during the administration of lobaplatin. In this study, one case of subcutaneous hemorrhage and one case of phlebitis were observed in the lobaplatin-based chemotherapy group, and significant improvement was observed after symptomatic treatment. Gastrointestinal toxicities occurred in both groups, with no significant difference between the groups. No obvious neurotoxicity or liver or kidney toxicity was observed in the two chemotherapy groups.

In summary, our study suggests that the addition of lobaplatin to the docetaxel and epirubicin regimen as neoadjuvant chemotherapy for TNBC significantly improves the TpCR rates and ORRs, with tolerable side effects; in addition, it can reduce recurrence and metastasis, at least in the short term. The TpCR rate can be used as a surrogate marker to evaluate short-term efficacy, but we do not currently have sufficient data to evaluate the long-term prognosis of TNBC patients. More analyses of follow-up information were limited by insufficient follow-up time. Subsequent results from ChiCTR-TRC-14005019 will be needed to determine whether the inclusion of lobaplatin leads to improvements in the long-term outcomes (the 5-year disease-free survival rate and 5-year overall survival rate) in TNBC and to identify molecular markers that profile treatment success.

## Methods

**Ethical statement**. This study was reviewed and approved by the Ethics Committee of the First Affiliated Hospital of Third Military Medical University, Chongqing, China, and was registered in the Chinese Clinical Trial Registry on May 29, 2014 (Registration Number ChiCTR-TRC-14005019). Written informed consent and a statement confirming consent to publish were obtained from the patients before their assignment to a treatment group.

**Patients**. This clinical trial was a prospective open-label RCT conducted at The First Affiliated Hospital of the Third Military Medical University.

The inclusion criteria were as follows: (1) patients with operable, previously untreated, clinical stage I to III (T1b-1c, N1-3, M0 or T2-4, N0-3, M0), ER- and PR-negative[39] (defined as ER and PR expression < 10% nuclei staining, according to the 2010 American Society of Clinical Oncology/College of American Pathologists guidelines), HER2-negative[12] (IHC 0-1 + or FISH ratio < 2.0), non-inflammatory invasive breast cancer confirmed by biopsy; (2) patients aged at most 70 years with a Karnofsky score of at least 70 points; (3) patients with adequate hematologic, renal, and hepatic function (defined as an absolute neutrophil count ≥ 1.5 × 10$^9$ l$^{-1}$, a hemoglobin count ≥ 80 g l$^{-1}$, a platelet count ≥ 75 × 10$^9$ l$^{-1}$, creatinine ≤ 1.5 × the upper limit of the normal range (ULN), total bilirubin ≤ 1.5 × ULN, or alanine aminotransferase or aspartate aminotransferase ≤ 3 × ULN)[40], normal cardiopulmonary and cardiac function as assessed by echocardiography, and no surgical indication or contraindication for chemoradiotherapy; and (4) women with childbearing potential who showed a negative result on a pregnancy test.

Patients with grade 2 or higher neuropathy and patients with contraindications for treatment with platinum, including uncontrolled thrombocytopenia or other malignant tumors, were excluded.

**Randomization and masking**. The randomized results were stratified according to the Karnofsky performance status (100, 90 or 80 vs. 70). Within each stratum, the patients were randomly assigned (1:1) to receive docetaxel and epirubicin (TE)

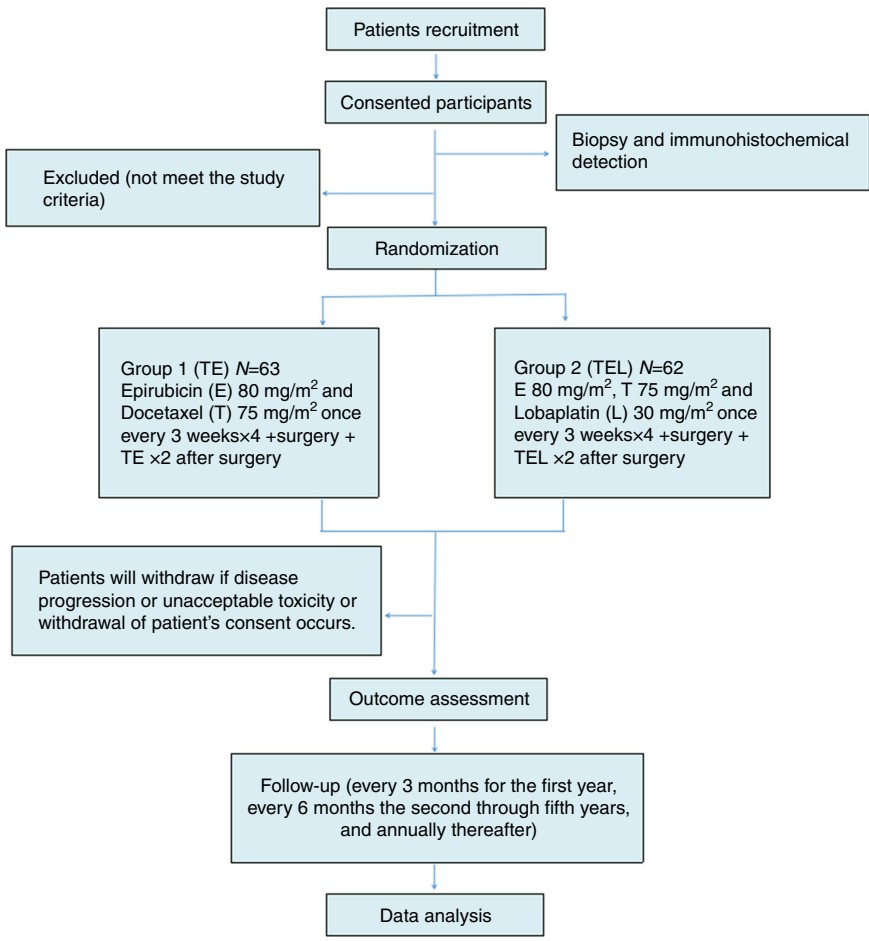

**Fig. 4** The trial flow chart. TE, docetaxel and epirubicin; TEL, docetaxel, epirubicin, and lobaplatin

with or without concurrent lobaplatin (L) treatment. The randomization sequence was generated with Research Randomizer (www.randomizer.org). The study's statistical team was blinded to the treatment assignments, but the patients and investigators were not blinded (open label).

**Treatment method**. Color ultrasonography was preferred for baseline breast imaging, and magnetic resonance imaging (brain, spine, bone, or vertebra) or computed tomography (chest, abdomen, spine, or bone) to rule out overt metastatic disease was recommended for patients with clinical stage III disease. Histologic confirmation by biopsy or coarse needle biopsy was encouraged for patients with clinically positive axillae. The surgeons were asked to assess each patient's eligibility for several surgical options, and tumor biopsies for correlative studies were required.

Figure 4 illustrates the treatment flow chart. Prior to surgery, all the patients received four cycles of 75 mg m$^{-2}$ docetaxel (T) and 80 mg m$^{-2}$ epirubicin (E) every 3 weeks with myeloid growth factor support, and the patients also received two cycles of this treatment after surgery. The patients were randomly assigned to receive TE with or without concurrent 30 mg m$^{-2}$ l. Radiotherapy was then performed for patients with indications according to the NCCN guidelines. The patients were given antiemetic drugs prior to chemotherapy. The new adjuvant chemotherapy treatment continued unless the occurrence of disease progression, intolerability, or withdrawal of consent. Toxicities were graded using the National Cancer Institute's Common Terminology Criteria for Adverse Events version 3.0[33], and toxicity tests were performed before and after the first day of chemotherapy and once a week during the administration of chemotherapy. If toxicity was observed, appropriate and timely interventions were undertaken. To ensure the continuity of chemotherapy, recombinant human granulocyte colony-stimulating factor was used in cases of grade 1+ leukopenia or neutropenia; in cases of grade 3 + anemia and thrombocytopenia, erythrocyte growth factors and interleukin-11 treatment was administered, respectively. In addition, if grade 3–4 toxicity was detected, the L doses could be withheld for up to 2 weeks and then resumed if the toxicity was resolved or reduced if the toxicity was not alleviated.

Lobaplatin was delayed for platelet counts <50,000/l and permanently reduced by 25% after a 2-week delay for thrombocytopenia or platelet counts <25,000/l at any time. The treatment was stopped if disease progression or unacceptable toxicity occurred or if the patient withdrew consent.

After the completion of neoadjuvant chemotherapy treatment (NACT), the patients underwent repeat mammary color ultrasonography to detect the effects of NACT followed by surgery. The extent of surgery and subsequent irradiation were determined by the treating physicians with consideration of the axillary lymph node metastasis status. Histopathology of the residual tumor was obtained for consenting patients.

**Follow-up**. Each patient was scheduled for follow-up visits every 3 months for the first year, every 6 months for the second to the fifth years, and annually thereafter. Each follow-up visit included a chest X-ray and abdominal and breast ultrasound. CT, MRI, or bone scan examination was necessary for patients with suspected metastatic lesions.

**Observational indexes**. Pathologic CR in the breast and the axilla (TpCR), the primary outcome measure, was defined as pathologic CR in the breast with the absence of any tumor deposit ≥ 0.2 mm in sampled axillary nodes or with negative pretreatment sentinel lymph nodes. Pathologic CR in the breast was regarded as the complete disappearance of residual invasive disease with or without ductal carcinoma in situ by histopathologic examination. TpCR is one of the evaluation criteria for NACT. Moreover, treatment responses were evaluated according to the WHO Response Evaluation Criteria in Solid Tumors: CR: visible lesions completely disappear on ultrasound after four cycles of NACT; PR: tumor size is reduced by more than 50% after four cycles of NACT; stable disease (SD): tumor size is reduced by less than 50% or increased by no more than 25%; and progression of disease (PD): tumor size is increased by more than 25%, or new lesions are formed. The ORR equals CR plus PR. Recurrence and metastasis were used as preliminary survival information. We defined recurrence as the reappearance of the carcinoma at the site of the surgical intervention and metastasis as any recurrence in lymph nodes or distant organs. Anticancer toxic drug reactions were graded according to the Common Terminology Criteria for Adverse Events, v3.0[33]. Dose-limiting toxicity (DLT) was defined as the occurrence of one or more of the following events after the first day of chemotherapy: (1) hematological toxicity in the form of grade 3–4 leukopenia, neutropenia, anemia and thrombocytopenia, (2) non-hematological toxicity with grade 3–4 vomiting, diarrhea, subcutaneous hemorrhage, phlebitis, liver and kidney toxicity and neurotoxicity, and (3) death.

The primary endpoint was the TpCR rate. The secondary endpoint was ORR and anticancer toxic drug reactions. In addition, the recurrence and metastasis rates were reported as preliminary follow-up information.

**Statistical analysis**. The data were analyzed using SPSS 23.0 (IBM Corp., Armonk, NY, USA) software. A $\chi^2$ test or Fisher's exact test was used to assess the differences between the experimental and control groups, and 95% CIs for the TpCR and ORR rates were calculated using binomial methods. One-sample Shapiro–Wilk tests and $t$-tests were used for the analysis of continuous variables for independent samples (such as age and tumor size). In addition, rank-sum tests were used for class distribution samples. The recurrences and metastases were analyzed, and Kaplan–Meier curves were plotted to assess the difference in preliminary survival information between the two treatment groups. Hazard ratios (HRs) were calculated for TE vs. TEL using univariate Cox proportional hazards regression models. Two-sided $P < 0.05$ were considered to indicate statistical significance.

**Data availability**. The data that support the findings of this study are available within the article and the Supplementary Files or from the authors upon request.

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

### Acknowledgements

We thank all the patients and the research staff for their contributions to this project. This study was financially supported by the Clinical Research Fund of Southwest Hospital, China (No. ChiCTR-TRC-14005019) and the Natural Science Fund of China (81302315).

### Author contributions

Y.Z. and X.J.W. conceived and designed the trial. X.J.W., P.T., and Y.Z. collected clinical data. X.J.W., S.F.L., T.Z., and L.Z. analyzed and interpreted the data. L.Z., S.S.W., L.R., and Y.Y.L. accomplished the screening of the enrolled patients; X.J.W. and Y.Z. wrote this paper. All authors finally approved the manuscript.

### Additional information

**Competing interests:** The authors declare no competing financial interests.

