## [Peer Review File · Nature Communications]

Reviewers' comments:

Reviewer #1 (Remarks to the Author):

Abstract, line 25. Adverse events were not similar in the 2 groups, TEL was associated with more haematological toxicity.

Lines 76-78. When and where was the study registered?

Lines 82-89. The study inclusion criteria are not clear. How are the stages I to III defined? How was HER2 assessed, and when was it considered positive? Why was the 10% cut-off considered for ER and PgR, and not the current recommendation of 1%? What does "adequate" haematologic, renal and hepatic function mean? How was the cardiac function assessed? Why do the authors refer to chemoradiotherapy, although this is not usually used in the treatment of early breast cancer?

Lines 107-113, and line 139. It is not clear who toxicity was assessed. Provide the laboratory tests that were performed, their intervals, and the exact reference for toxicity grading. Was toxicity collected prospectively on case report forms, or retrospectively from the case records?

Lines 109-113. Why were dose reductions done based on thrombocytopenia only?

There is no description about locoregional radiotherapy. Was it given, and how?

Lines 127-138. Clarify what pCR means, does it refer to pCR in the breast only, or both the breast and the axillary nodes?

Lines 140-143. The power calculations for the sample size are not provided. Were the continuous variables, such as age and tumour size, normally distributed to allow the use of the t test?

Line 150. How was randomisation carried out? Provide the details in the Methods section. Was there any stratification involved?

Lines 178-180. Provide the P values for the difference in haematological toxicity between the groups.

Line 178 and Table 3. The frequency of grade 3-4 anaemia in the TEL group (52.5%) is surprisingly high considering the short duration of the treatment, suggesting that lobaplatin is very myelotoxic. Define what grade III and IV anaemia mean. How many patients received red cell infusions or erythrocyte growth factors? Discuss the rate of Grade 3-4 anaemia in Discussion and compare this figure with a few commonly used neoadjuvant regimens.

Discussion, lines 196-320. I find Discussion lengthy and somewhat biased. It contains comparisons between series that seem not directly comparable, and discussion of observations that are based on very small numbers (e.g. lines 275-291).

Table 1. Provide information about the primary tumor size, histological type and histological grade.

Minor:

p. 1. Indicate in the title that the study is a randomized study.

Provide the numbers that the percentages correspond to throughout the manuscript.

Lines 46-48. Clarify this sentence.

Lines 68-69. Lobaplatin has probably been approved for these indications in China, not by the FDA or the EMA.

Line 95. Provide the sites imaged with MRI or CT.

Line 114. "New" should probably be replaced with "neo".

Lines 130 and 160. These sentences are almost identical.

Line 144. Are all P values 2-sided?

Line 153, Table 1. Provide more data about the primary tumor size.

Lines 160-169. Provide the 95% confidence intervals for the response rates.

Table 3. Provide the P values for haematological toxicity between the groups. Provide also the data for Grade 1 and 2 haematological side effects.

Table 4. It is unclear how liver and kidney toxicity were evaluated.

Reviewer #2 (Remarks to the Author):

The report is fairly straight forward, but would benefit from some additional work. A few points that must be addressed:

- 1) There are several parts of the paper that would benefit from edits as it relates to grammar and syntax.
- 2) Some of the discussion in the discussion would be better at the introduction...ie the rationale for lobaplatin and description of its preclinical data supporting its use.
- 3) There are parts of the paper that refer to other trials and improved outcome with the use of platinum but the actual data is not provided, only generalizations about benefit.
- 4) an explicit definition of pCR should be discussed because it varies between trials. ...ie did they include pcr in nodes as well?...some trials do
- 4) they have no real long-term data so would not even describe as it is meaningless.

Respond to the reviewers

Reviewers' comments:

Reviewer #1 (Remarks to the Author):

Abstract, line 25. Adverse events were not similar in the 2 groups, TEL was associated with more haematological toxicity.

Reply: We have corrected this in the abstract of the revised article.

Lines 76-78. When and where was the study registered?

Reply: This study was registered in Chinese Clinical Trial Registry in May 29, 2014. The registration number is ChiCTR-TRC-14005019. This information is provided in the revised article.

Lines 82-89. The study inclusion criteria are not clear. How are the stages I to III defined?

Reply: The study inclusion criteria have been provided in detail in the revised article.

How was HER2 assessed, and when was it considered positive? Why was the 10% cut-off considered for ER and PgR, and not the current recommendation of 1%?

Reply: We have clarified this information in the revised article. The disease was considered HER2-positive disease if the tumor tissue shows a fluorescence *in situ* hybridization (FISH) ratio of at least 2.0 or

immunohistochemistry (IHC) 3+. In contrast, HER2 negativity was defined as IHC 0-1+ or FISH ratio<2.0. Histologically or cytologically confirmed estrogen receptor-negative and progesterone receptor-negative disease was determined through a local pathology immunohistochemistry assessment (defined by the 2010 American Society of Clinical Oncology/College of American Pathologists guidelines as <10% nuclei staining). In addition, the characteristics of breast cancer patients with ER and PR expression less than 10% were similar to those of patients with ER and PR expression less than 1% during our clinical treatment and follow-up. Therefore, a cut-off of 10% rather than 1% was considered for ER and PR in the current trial.

What does “adequate” haematologic, renal and hepatic function mean?

Reply: We have clarified this information in the revised article.

How was the cardiac function assessed? Why do the authors refer to chemoradiotherapy, although this is not usually used in the treatment of early breast cancer?

Reply: Cardiac function was assessed by echocardiography. Adjuvant chemotherapy is required for TNBC patients with a maximal tumor diameter greater than 0.5 cm or axillary lymph node metastasis according to the National Comprehensive Cancer Network (NCCN) guidelines. In the current trial, the maximum diameter of the tumor in every patient exceeded 0.5 cm.

Lines 107-113, and line 139. It is not clear who toxicity was assessed. Provide the laboratory tests that were performed, their intervals, and the exact reference for toxicity grading. Was toxicity collected prospectively on case report forms, or retrospectively from the case records?

Reply: We have clarified this information in the revised article. Toxicity was prospectively collected on case report forms.

Lines 109-113. Why were dose reductions done based on thrombocytopenia only?

Reply: The reason has been discussed in detail in the Discussion section of the revised article.

There is no description about locoregional radiotherapy. Was it given, and how?

Reply: We have described this information in the revised article. Radiotherapy was performed for patient with indications after surgery and chemotherapy in accordance with the NCCN guidelines.

Lines 127-138. Clarify what pCR means, does it refer to pCR in the breast only, or both the breast and the axillary nodes?

Reply: pCR has been defined in the revised article. Pathologic complete response in the breast (BpCR) was defined as the complete disappearance of residual invasive disease with or without ductal carcinoma *in situ*, as determined by histopathologic examination. pCR breast/axilla (TpCR) was defined as BpCR with the absence of any tumor deposit ≥ 0.2 mm in

sampled axillary nodes or with negative pretreatment sentinel lymph nodes. In the current trial, pCR refers to BpCR.

Lines 140-143. The power calculations for the sample size are not provided. Were the continuous variables, such as age and tumour size, normally distributed to allow the use of the t test?

Reply: We have revised this information in the revised article. One-sample Shapiro-Wilk tests show that the age ($P=0.230$ in TE, $P=0.713$ in TEL) and tumor size ($P=0.084$ in TE, $P=0.077$ in TEL) showed a normal distribution. In addition, the age ($t=1.878$, two-sided $P=0.063$) and tumor size ($t=0.508$, two-sided $P=0.612$) were evenly distributed between the two groups.

Line 150. How was randomisation carried out? Provide the details in the Methods section. Was there any stratification involved?

Reply: The details about the randomization are provided in the Methods section of the revised article.

Lines 178-180. Provide the P values for the difference in haematological toxicity between the groups.

Reply: The P values are provided in the Results section of the revised article.

Line 178 and Table 3. The frequency of grade 3-4 anaemia in the TEL group (52.5%) is surprisingly high considering the short duration of the treatment, suggesting that lobaplatin is very myelotoxic. Define what

grade III and IV anaemia mean. How many patients received red cell infusions or erythrocyte growth factors? Discuss the rate of Grade 3-4 anaemia in Discussion and compare this figure with a few commonly used neoadjuvant regimens.

Reply: We have clarified this information in the Methods and Discussion sections of the revised article.

Discussion, lines 196-320. I find Discussion lengthy and somewhat biased. It contains comparisons between series that seem not directly comparable, and discussion of observations that are based on very small numbers (e.g., lines 275-291).

Reply: We have revised the Discussion section in the revised article.

Table 1. Provide information about the primary tumor size, histological type and histological grade.

Reply: In the revised article, information of the primary tumor size, histological type and clinical stage are provided in Table 1, and the histology of all of the patient enrolled in this trial revealed non-inflammatory invasive breast cancer.

Minor:

p. 1. Indicate in the title that the study is a randomized study. Provide the numbers that the percentages correspond to throughout the manuscript.

Reply: We have revised the information in the revised article.

Lines 46-48. Clarify this sentence.

Reply: We have clarified this sentence in the revised article.

Lines 68-69. Lobaplatin has probably been approved for these indications in China, not by the FDA or the EMA.

Reply: We made the corresponding correction in the revised article.

Line 95. Provide the sites imaged with MRI or CT.

Reply: The sites imaged by MRI are the brain, spine, bone or vertebra, and the sites imaged by CT are the chest, abdomen, spine or bone.

Line 114. “New” should probably be replaced with “neo”.

Reply: We have made the corresponding correction in the revised article.

Lines 130 and 160. These sentences are almost identical.

Reply: The repeated sentences in lines 160 have been deleted.

Line 144. Are all P values 2-sided?

Reply: Yes, all P values are two-sided.

Line 153, Table 1. Provide more data about the primary tumor size.

Reply: Data of the primary tumor size are provided in Table 1 in the revised article.

Lines 160-169. Provide the 95% confidence intervals for the response rates.

Reply: This information is provided in the revised article.

Table 3. Provide the P values for haematological toxicity between the groups. Provide also the data for Grade 1 and 2 haematological side effects.

Reply: In the revised article, the P values for grade 3-4 hematological toxicity are provided in the Results section, and data for grade 1-2 hematological side effects are provided in Table 3.

Table 4. It is unclear how liver and kidney toxicity were evaluated.

Reply: Grade III-IV treatment-related liver and kidney toxicity were evaluated, and the toxicities were graded using the National Cancer Institute's Common Terminology Criteria for Adverse Events version 3.0.

Reviewer #2 (Remarks to the Author):

The report is fairly straight forward, but would benefit from some additional work. A few points that must be addressed:

1) There are several parts of the paper that would benefit from edits as it relates to grammar and syntax.

Reply: The language in the revised article has been revised by a professional organization (AJE).

2) Some of the discussion in the discussion would be better at the introduction...ie the rationale for lobaplatin and description of its preclinical data supporting its use.

Reply: We made corrections in the revised article according to your

comments.

3) There are parts of the paper that refer to other trials and improved outcome with the use of platinum but the actual data is not provided, only generalizations about benefit.

Reply: The actual data are included in the revised article.

4) an explicit definition of pCR should be discussed because it varies between trials...ie did they include pcr in nodes as well?...some trials do

Reply: pCR is defined in detail in the revised article. Pathologic complete response in the breast (BpCR) was defined as the complete disappearance of residual invasive disease with or without ductal carcinoma *in situ*, as determined by histopathologic examination. pCR breast/axilla (TpCR) was defined as BpCR with the absence of any tumor deposit ≥ 0.2 mm in sampled axillary nodes or with negative pretreatment sentinel lymph nodes. In the current trial, pCR refers to BpCR.

4) they have no real long-term data so would not even describe as it is meaningless.

Reply: Long-term outcomes, including recurrence-free (RFS) and overall survival (OS), are effective indexes for evaluating the therapeutic efficacy of TNBC patients. A meta-analysis published in Lancet in 2014 confirmed that patients with TNBC who achieved a pCR exhibited superior event-free survival (HR, 0.24; 95% CI, 0.18 - 0.33) and OS (HR, 0.16; 95% CI, 0.11 - 0.25) than those who did not achieve a pCR.

Therefore, the short-term assessment (PCR and ORR) is necessary and valuable. Subsequent results from ChiCTR-TRC-14005019 will be necessary to determine whether the inclusion of lobaplatin leads to improvements in long-term outcomes in TNBC and to identify molecular markers that profile treatment success. So, follow-up is described in the article.

Reviewers' comments:

Reviewer #1 (Remarks to the Author):

1. The overall presentation still needs improvement. There are grammar mistakes, and sentences that are not clear (e.g. Abstract, lines 13-14 and lines 19-20; p. 2, lines 41-43; p. 4, lines 76-77; line 85; p. 13, lines 281-282; p. 16, lines 330-332).
2. The primary end point is not defined. The authors provide data about the response rate (ORR) and the pCR rate in the breast (BpCR), but not about the pCR rate in the breast and the axilla, which is often considered a more relevant end point than the pCR rate in the breast
3. The lobaplatin regimen (TEL) had substantial bone marrow toxicity despite extensive supporting therapy (granulocyte growth factor, erythropoietin, and IL-11 support). The experimental arm (TEL) was more myelotoxic than the comparator arm (TE), and it remains unknown whether a higher response rate would have been achieved in the control group if higher, but equally toxic doses had been administered in the TE group. This is not discussed. It may not be accurate to state that TEL had mild toxicity (p. 7, line 153).
4. The study protocol was not included. The inclusion criteria (p. 14, lines 298-299) discuss metastatic liver disease, which seems odd in a study protocol on early breast cancer.
5. p. 7, lines 142-148. The survival data seem scarce and are not based on life-table analyses.
6. Discussion is still lengthy, and appears to contain sentences that may not be relevant to the manuscript (e.g. p. 10, lines 209-213) or that may suit better to Materials & Methods (p. 11, lines 223-228).

Minor:

1. Abstract, line 92. Explain abbreviations BpCR and ORR.
2. Abstract, line 15. The treatment may have failed, not the patients.
3. The median size of the breast tumors and the size range are not provided. These are relevant, as these might influence the pCR rate.
4. p. 2, line 31-32, Discussion: Some drug targets have been identified in TNBC, such as DNA repair gene aberrations for PARP inhibition.
5. Provide the references that find pCR as the most powerful prognostic factor in TNBC.

Reviewer #2 (Remarks to the Author):

The extensive revisions in response to the reviewers critique now position this paper for publication !

Responses to the reviewers

Reviewers' comments:

Reviewer #1 (Remarks to the Author):

1. The overall presentation still needs improvement. There are grammar mistakes, and sentences that are not clear (e.g., Abstract, lines 13-14 and lines 19-20; p. 2, lines 41-43; p. 4, lines 76-77; line 85; p. 13, lines 281-282; p. 16, lines 330-332).

Reply: The language in the revised article has been revised by a professional organization (AJE) again.

2. The primary end point is not defined. The authors provide data about the response rate (ORR) and the pCR rate in the breast (BpCR), but not about the pCR rate in the breast and the axilla, which is often considered a more relevant end point than the pCR rate in the breast

Reply: We have corrected the information and data associated with pCR in the revised article. The primary end point is defined in the Materials and Methods section of the revised article. Pathologic complete response in the breast and the axilla (TpCR), the primary outcome measure, was defined as pathologic complete response in the breast with the absence of any tumor deposit ≥ 0.2 mm in sampled axillary nodes or with negative pretreatment sentinel lymph nodes. Pathologic complete response in the breast was regarded as the complete disappearance of residual invasive disease with or without ductal carcinoma in situ upon histopathologic examination.

3. The lobaplatin regimen (TEL) had substantial bone marrow toxicity despite extensive supporting therapy (granulocyte growth factor, erythropoietin, and

IL-11 support). The experimental arm (TEL) was more myelotoxic than the comparator arm (TE), and it remains unknown whether a higher response rate would have been achieved in the control group if higher, but equally toxic doses had been administered in the TE group. This is not discussed. It may not be accurate to state that TEL had mild toxicity (p. 7, line 153).

Reply: The relevant content has been discussed in detail in the Discussion section of the revised article. In one phase III trial, the time to disease progression or overall survival for docetaxel doses ranging from 60 to 100 mg m⁻² was similar for the adjuvant treatment of intention-to-treat patients with advanced breast cancer; however, significantly aggravated hematologic toxicity occurred in the highest docetaxel dose group, which had a 93.4% incidence of grade 3 to 4 neutropenia²². In addition, in an analysis of the FinHer trial data, there were no significant differences in the baseline characteristics of the patients or the tumors, the numbers of distant recurrences or deaths between the groups receiving 80 mg m⁻² and 100 mg m⁻² of docetaxel²⁴. In addition, in our clinical trial, the incidence of grade III-IV myelotoxicity was not significantly different between patients with pCR and without pCR. Therefore, in the control group, when the drug dosage increased, there was more myelotoxicity without necessarily a higher response rate.

4. The study protocol was not included. The inclusion criteria (p. 14, lines 298-299) discuss metastatic liver disease, which seems odd in a study protocol on early breast cancer.

Reply: We have corrected the study protocol and describe it in detail in

the Methods section and Figure 1 of the revised article.

5. p. 7, lines 142-148. The survival data seem scarce and are not based on life-table analyses.

Reply: Because the early trial was recently completed, the follow-up times are not extensive at present. We have reported the recurrence and metastasis rates as preliminary survival information in the revised article. Additional analyses of follow-up information were limited by insufficient follow-up time. Subsequent results from ChiCTR-TRC-14005019 will be needed to determine whether the inclusion of lobaplatin leads to improvements in the long-term outcomes (5-year disease-free survival rate and 5-year overall survival rate) of TNBC and to identify molecular markers that predict treatment success.

A meta-analysis published in Lancet in 2014 confirmed that patients with TNBC who achieved pCR exhibited superior event-free survival (HR, 0.24; 95% CI, 0.18 - 0.33) and OS (HR, 0.16; 95% CI, 0.11 - 0.25) compared with those who did not achieve pCR. Therefore, the short-term assessment (pCR and ORR) is necessary and valuable, and the pCR rate can be used as a predictive marker to evaluate long-term efficacy.

6. Discussion is still lengthy, and appears to contain sentences that may not be relevant to the manuscript (e.g., p. 10, lines 209-213) or that may suit better to Materials & Methods (p. 11, lines 223-228).

Reply: We have revised the Discussion section in the revised article.

Minor:

1. Abstract, line 92. Explain abbreviations BpCR and ORR.

Reply: We have added explanations in the revised article.

2. Abstract, line 15. The treatment may have failed, not the patients.

Reply: We have changed this statement in the revised article.

3. The median size of the breast tumors and the size range are not provided. These are relevant, as these might influence the pCR rate.

Reply: The median size of the breast tumors and the size range are provided in the Results section in the revised article.

4. p. 2, line 31-32, Discussion: Some drug targets have been identified in TNBC, such as DNA repair gene aberrations for PARP inhibition.

Reply: We have corrected this information in the revised article.

5. Provide the references that find pCR as the most powerful prognostic factor in TNBC.

Reply: A meta-analysis published in Lancet in 2014 confirmed that patients with TNBC who achieved pCR exhibited superior event-free survival (HR, 0.24; 95% CI, 0.18 - 0.33) and OS (HR, 0.16; 95% CI, 0.11 - 0.25) compared with those who did not achieve pCR and that pCR is the most powerful prognostic factor in TNBC²⁶. Another study that compared the response to neoadjuvant chemotherapy and survival between patients with and without TNBC found that patients with TNBC have increased pCR rates compared with non-TNBC patients and that those with pCR have excellent survival³. These studies have been used as references in our article.

Reviewer #2 (Remarks to the Author):

The extensive revisions in response to the reviewers critique now position this paper for publication!

REVIEWERS' COMMENTS:

Reviewer #2 (Remarks to the Author):

I think the issues outlined by reviewers have now been sufficiently and adequately addressed by the authors and as such the manuscript is appropriate for publication.